**Data availability statement:** Database Url: https://fcon_1000.projects.nitrc.org/indi/indiPRIME.html DOI: 10.1016/j.neuron.2018.08.039.

# RISNet: A variable multi-modal image feature fusion adversarial neural network for generating specific dMRI images

Guolan Wang[1], Xiaohong Xue[1], Yifei Chen[2], Hao Liu[2], Haifang Li[1,2]*, Qianshan Wang[3]*

**1** College of Computer and Information Engineering, Shanxi Technology and Business University, Taiyuan, China, **2** College of Computer Science and Technology (College of Data Science), Taiyuan University of Technology, Taiyuan, China, **3** College of software, Taiyuan University of Technology, Taiyuan, China

* lihaifang@tyut.edu.cn (H.L.); wangqianshan0203@gmail.com (Q.W.)

## Abstract

The b-value in the diffusion magnetic resonance image(dMRI) reflects the degree to which the water molecules are affected by the magnetic field gradient pulse in the tissue, and the different b-values not only affect the image contrast but also the accuracy of the subsequent calculation. The imbalance between the lower and higher b-value image categories in the macaque dMRI brain imaging dataset dramatically affects the accuracy of computational neuroscience. The medical image conversion method based on the generative adversarial network can generate different b-value images. However, the macaque brain dataset has multi-center and small-sample problems, which restricts the training effect of the general model. To increase macaques' lower b-value dMRI data, we propose a variable multi-modal image feature fusion adversarial neural network called RISNet. The network can use the proposed rapid insertion structural(RIS) to input features from different modes into a general residual decoding structure to enhance the model's generalization ability. The RIS combines the advantages of multi-modal data, which can quickly rewrite the network and extract and fuse the features of multi-modal data. We used a T1 image and a higher b-value image of the brain as model inputs to generate high-quality, lower b-value images. Experimental results show that our method improves the PSNR index by 1.8211 on average and the SSIM index by 0.0111 compared with other methods. In addition, in terms of qualitative observation and DTI estimation, our process also shows sound visual effects and strong generalization ability. These advantages make our method an effective means to solve the problem of dMRI brain image conversion in macaques and provide strong support for future neuroscience research.

## Introduction

As non-human primates, macaques have become a key model in neuroscience research due to their genetic and anatomical similarities with humans. In particular, in terms of exploring the brain's structure, function, and disease mechanisms, the results of macaque research

**Funding:** This work was supported by Scientific and Technological Innovation Programs of Higher Education Institutions in Shanxi (2023L494 to G.W.), the National Natural Science Foundation of China (61976150 to H.L.), and Shanxi Provincial Natural Science Foundation Youth Fund Project (2021030212427 to Q.W.). The funders had no role in study design, data collection and analysis, decision to publish, or preparation of the manuscript.

**Competing interests:** The authors have declared that no competing interests exist.

are of great value for understanding the working mechanism of the human brain. Therefore, accurate and efficient analysis of brain imaging data of macaques is of great significance to promote the development of neuroscience [1,2].

Using diffusion magnetic resonance imaging (dMRI), as a non-invasive medical imaging technique, can delineate the fine structure of the white matter fiber tract of the brain, providing an unprecedented perspective for neuroscience research [3,4]. The b-value is a key parameter during dMRI acquisition that quantifies the intensity of the influence of the diffusion-sensitive gradient field applied to the tissue [5]. The b-value determines the sensitivity of the image to the diffusion process of water molecules. A higher b-value means greater diffusion sensitivity and can more effectively detect the influence of the internal structure of the tissue on the diffusion path of water molecules, which can help reveal microstructural features such as nerve fiber bundles [6]. However, as the b-value increases, the signal-to-noise ratio (SNR) decreases, as the signal difference is more significant when the conditions are higher than those that do not diffuse freely.

Different b-values need to be set during dMRI acquisition. Typically, one lower b-value image corresponds to multiple higher b-value images, and this ratio can ideally reach 5-10 times [7,8]. The acquisition and processing of different b-value images in dMRI is essential for subsequent fiber tract tracing and structural analysis. However, due to the differences in early acquisition protocols and significant differences in the parameters of scanning equipment at different centers, various factors affect the data quality of publicly available macaque dMRI images data quality. The number of images with low b-value is often insufficient or even missing, which poses a challenge for further neurocomputing [9].

In recent years, generative adversarial networks (GAN) have made remarkable progress in the field of image generation and conversion [10,11], which provides new ideas for solving this problem [12–14]. GAN can learn the features and distribution of low-b-value images from existing modal images and generate high-quality low-b-value images. This method can not only solve the problem of the insufficient number of low-b-value images in the data set but also improve the quality and information integrity of dMRI images and provide a more reliable data basis for subsequent fiber bundle tracing and structural analysis.

At present, GAN has been widely used in the field of medical image conversion. The Synb0-DisCo method applied the pix2pix method to correct the distorted b0 image [15–17]. pGAN uses the residual network as the generator architecture and introduces VGG loss [18, 19]. EA-GAN starts from the edge information to improve the texture details of the generated images [20]. MedGAN uses cascaded U-Net as its generator for a variety of medical image conversion tasks [21]. SwinUnet converts magnetic resonance brain images based on Transformer, which is applied to the field of medical image conversion [22,23]. ResViT combines attention mechanisms and convolutional networks to improve image quality further [24,25].

Although GAN has a wide range of applications in image conversion, most medical brain image conversion methods are based on human brain image data. Compared with the human brain image, the macaque images at different centers are quite different; the sample size is small, and the brain structure of the macaque is different from that of the human brain [26]. The multi-center joint training method can increase the data sample size. Still, applying the method based on human brain image transformation to multi-center joint training may have the problem of insufficient generalization ability. In addition, existing conversion methods are often limited to uni-modal conversion; that is, only a single type of medical imaging data is used for conversion. Multi-modal medical imaging data usually provides more comprehensive and accurate brain information in neuroscience. Therefore, multi-modal data combined with multi-center data training can expand the training samples and enhance the ability to generate the neural network model [27]. However, how to ensure that the features extracted

from the data of different modalities can be fused and not produce excessive bias to a specific modality [28].

To solve the above problems, this paper proposes a rapid insertion structure(RIS) to combine the advantages of multi-modal data based on residual decoders. Our innovative residual decoder has powerful feature extraction and expression capabilities and realizes the accurate conversion from high-b-value images and T1 images to low-b-value images. Through this method, we can generate high-quality images to make up for the insufficient number of low-b-value images in the dMRI data and make full use of multi-modal medical imaging information to improve the accuracy and reliability of macaque brain image analysis. The specific contributions of this article are as follows:

This paper introduces a Rapid Insertion Structure (RIS) combined with residual decoders to the issues above by integrating the benefits of multi-modal data using the GAN method. The novel residual decoder employed in this paper exhibits robust feature extraction and expression capabilities, facilitating precise conversion from high-b-value images and T1 images to low-b-value images. This approach enables the generation of high-quality images to compensate for the limited quantity of low-b-value images in dMRI data, leveraging the full potential of multi-modal medical imaging data to enhance the precision and dependability of macaque brain image analysis. The key contributions of this study are outlined as follows:

- Innovative Architecture for Multi-Modal Integration: we innovatively proposed a rapid insertion structure that requires only simple modifications to the model to extract and integrate various modal data. This mode can be rewritten effortlessly to combine features from various data types. This approach enables the application of multi-center joint training in expanding the macaque dMRI dataset during the experiment.
- Residual Decoder-Based Multi-Modal Fusion: A novel residual decoder framework is proposed for multi-modal dMRI image conversion. The residual decoder carries out multi-modal fusion and decoder feature fusion, which avoids the loss of information caused by information redundancy, makes full use of multi-modal information, and generates a more accurate diffusion MRI.

## Materials and methods

### Dataset

The PRIMatE Data Exchange (PRIME-DE) data center has made a rich dataset of primate brain images public [29]. We selected four datasets, including dMRI images from PRIME-DE provided by Aix-Marseille Universite (AMU), Mount Sinai School of Medicine-Siemens, University of California, Davis(UCDavis), and the University of Wisconsin-Madison(UWM). These four sites provide dMRI, fMRI, and T1-weighted images, offering a multi-modal dataset resource for our research. Detailed acquisition parameters for the dMRI images can be found in Table 1, and the T1-weighted images are listed in Table 2.

**Table 1**. Dataset partitioning.

| Dataset | Samples | Training | Test |
|---|---|---|---|
| AMU | 4 | 3 | 1 |
| MountSinai-S | 5 | 3 | 2 |
| UCDavis | 19 | 15 | 4 |
| UWM | 66 | 53 | 13 |

**Table 2**. dMRI acquisition parameters.

| Datasets | Scanner | Resolution(mm) | TE(ms) | TR(ms) | b-Values(s/mm2) |
|---|---|---|---|---|---|
| AMU | Siemens Prisma | $1.0 \times 1.0 \times 1.0$ | 87.6 | 7520 | 5;500 |
| MountSinai | Siemens Skyra | $1.0 \times 1.0 \times 1.0$ | 95 | 5000 | 10;1005 |
| UCDavis | Siemens Skyra | $1.4 \times 1.4 \times 1.4$ | 115 | 6400 | 5;1600 |
| UWM | GE DISCOVERY_MR750 | $2.2 \times 3.1 \times 2.2$ | 94.3 | 6100 | 0;1000 |

## Preprocessing

This study preprocesses dMRI images and T1-weighted images separately. The specific steps are as follows:

- Perform head motion and eddy current correction on dMRI images and T1 image data using the eddy tool in FSL software [30,31].
- Remove non-brain tissue from T1-weighted images and dMRI images using the brain tissue segmentation tool HC-Net [32].
- Extract paired high-b-value($b \geq 1000$) images and low-b-value($b = 0$) images from dMRI images using FSL tools. We combine the T1-weighted images and high-b-value images served as input for the model, and the low-b-value images serve as the ground truth.
- Normalize all pixel values of the images to the range of 0 to 1 using the maximum-minimum normalization method.
- Resample all images to $256 \times 256 \times 256$ and slice all images into 2D slices in the coronal plane.

    Finally, the 2D slices of T1-weighted images and high-b-value images serve as input for the multi-modal generator. In contrast, the 2D slices of low-b-value images serve as input for the discriminator. We use multi-center data for joint model training to enhance model generalization and increase data diversity. We divide all data within each site into training and testing sets. The specific partition results are shown in Table 3.

## Overall framework design

During the model training process, T1, B0, and high-B-value magnetic resonance imaging (MRI) data are utilized as input. Prior to feeding the data into the model, downsampling is performed, and the original dimensions of the data are meticulously recorded. Subsequently, the resampled T1 and high-B-value data are fed into the generator. The generated B0_ data from the generator, along with the resampled real B0 data, are then used as inputs for the

**Table 3**. T1 acquisition parameters.

| Datasets | Scanner | Resolution(mm) | TE(ms) | TR(ms) | b-Values(s/mm2) |
|---|---|---|---|---|---|
| AMU | Siemens Prisma | $0.8 \times 0.8 \times 0.8$ | 2.04 | 2900 | 1000 |
| MountSinai | Siemens Skyra | $0.5 \times 0.5 \times 0.5$ | 3.02 | 2700 | 800 |
| UCDavis | Siemens Skyra | $0.3 \times 0.3 \times 0.3$ | 3.65 | 2500 | 1100 |
| UWM | GE DISCOVERY_MR750 | $0.3 \times 0.5 \times 0.3$ | 5.412 | 11.4 | 600 |

discriminator. The output of the generator undergoes concatenation and upsampling operations to restore it to the same dimensions as the original real data. Structural Similarity Index (SSIM), Peak Signal-to-Noise Ratio (PSNR), and Information Entropy (IM) are employed as evaluation metrics to assess the model's performance after each training epoch. The model with the optimal performance is ultimately selected as the final result.The overall system architecture diagram is shown in Figure A of Fig 1.

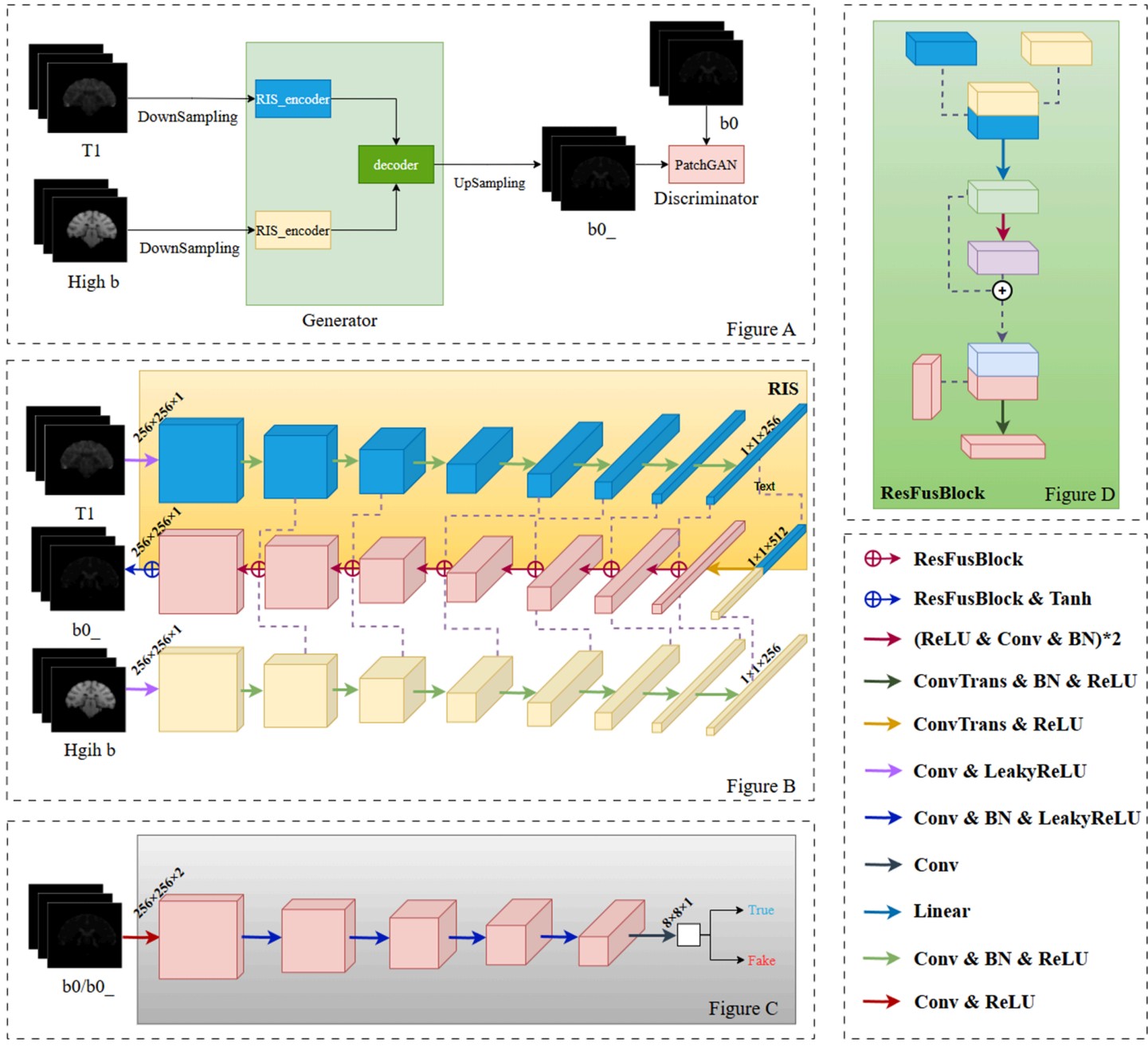

**Fig 1. Network architecture.**

## Generative adversarial network based on residual decoders

Our team proposes a generative adversarial network (GAN) that utilizes residual decoders to manage multi-modal data effectively. The most significant advantage of RIS lies in its non - interference with pre - trained models. All features are processed into an average feature vector before the network starts learning. As a result, when multimodal data is incorporated, there is no need to retrain the pre - trained model. Consequently, data from other modalities can be added for training at any time.The generator's architecture features two rapid insertion structural (RIS) encoders designed to process multi-modal data and a shared residual decoder. The discriminator employs a classical PatchGAN network to enhance the model's discriminative capabilities.

For data input, both T1 and high b-value images are fed simultaneously into two distinct RIS encoders. Each encoder processes its respective model data, resulting in 16 multi-scale feature maps.

The residual decoder then integrates information from these 16 multi-scale feature maps through a series of complex fusion operations. This integration allows the model to fully utilize the complementary aspects of the multi-modal data, thereby generating more accurate and realistic low-b-value images.

Ultimately, after being processed by the residual decoder and output through the Tanh activation function, the desired low-b-value image is produced. This entire process takes full advantage of the residual decoder, not only improving the quality of the generated images but also enhancing the stability and robustness of the model. Figure B of Fig 1 shows the design details.

**Encoder.** T1 images and high-b-value images are used as multi-modal data inputs, represented by $IN \in R^{B \times C \times H \times W}$. The data from different modalities enters the corresponding encoder. In each encoder, the data is first processed through convolutional blocks and the LeakyReLU activation function, as shown in Eq (1):

$$f_1 \in \mathbb{R}^{B \times 64 \times \frac{H}{2} \times \frac{W}{2}} = \text{LReLU}(\text{Conv}(\text{IN})) \tag{1}$$

where *LReLU* represents the LeakyReLU activation function, which is used to increase the nonlinear characteristics, and *Conv* represents the convolution operation, with a convolution kernel size of $3 \times 3$, which is used to extract the features of the image. $f_1 \in R^{B \times C \times H \times W}$ represents the first feature map obtained as the starting point for subsequent operations.

The feature maps are then processed by six identical downsampled convolution blocks. Each block performs convolution, batch normalization, and downsampling operations on the input feature map to generate feature representations at various scales. The output from each downsampled convolution block serves as the input for the subsequent block, creating a hierarchical mechanism for feature extraction, as demonstrated in Eq (2):

$$f_i \in \mathbb{R}^{B \times C^i \times H^i \times W^i} = \text{LReLU}(\text{BN}(\text{Conv}(f_{i-1}))), \quad i = 2, 3, \ldots, 7 \tag{2}$$

where *BN* indicates a batch normalization operation. $f_i \in R^{B \times C^i \times H^i \times W^i}$ represents the first *i* feature map, and at each stage, the size of the feature map gradually decreases. At the same time, the number of channels gradually increases to capture image information at different scales. Specifically, the size of H and W is reduced to 1/2 of their original size after each downsampling. At the same time, the number of channels C gradually increases from 64 to 512 depending on the depth of the convoluted block, and then it remains the same. Finally, the last feature map is obtained through the convolution block and the ReLU activation function. As

shown in Eq (3):

$$f_8 \in \mathbb{R}^{B \times 512 \times 1 \times 1} = \text{ReLU}(\text{Conv}(f_7)) \tag{3}$$

where *ReLU* indicates the *ReLU* activation function.

After processing this series of downsampled convolution blocks, each encoder obtained several feature maps of different scales. These feature maps not only contain the rich details of the image but also reveal its internal structure across various scales. By fully extracting the multi-scale features of multi-modal data, our method offers robust support for subsequent image generation and detail enhancement.

It's worth noting that our encoders are designed to be flexible and scalable. If the dataset contains more modal images, we only need to add the corresponding encoder path to extract the new multi-modal information. This design allows our method to accommodate datasets of different sizes and diversity to support multi-modal image generation tasks. Our encoders are designed to be both flexible and scalable. If the dataset includes additional modal images, we need to add the appropriate RIS encoder path to extract the new multi-modal information. This design enables our method to effectively accommodate datasets of varying sizes and diversity, supporting multi-modal image generation tasks and directly utilizing single-modal data for generation.

**Residual decoder.** The residual decoder is a core component of the proposed method, designed to efficiently fuse key information from multi-modal data and generate high-quality images. The decoder consists of eight residual fusion modules, each responsible for fusing two multi-scale feature maps from different encoders [33].

In the fusion process, each residual fusion module makes full use of the multi-modal feature map from the two encoders as well as the feature map of the previous residual fusion module. The first residual fusion module only fuses multi-modal feature maps to ensure the completeness and accuracy of the initial information. As the decoding process progresses, subsequent modules gradually incorporate more contextual information to enrich the details of the generated images. It is shown in Eq (4):

$$f_i^D \in \mathbb{R}^{B \times C^i \times H^i \times W^i} = \text{ResFusBlock}(f_i^{E1}, f_i^{E2}, f_{i-1}^D), \quad i = 1, 2, \dots, 8 \tag{4}$$

where $f_i^D \in R^{B \times C^i \times H^i \times W^i}$ represents the $i$ feature map of the decoder, *ResFusBlock*($\cdot$) represents the residual fusion module, $f_i^{E1}$ represents the $i$ feature map of the first encoder, and $f_i^{E2}$ represents the $i$ feature map of the second encoder.

To more fully fuse multi-modal information, we introduce a phased fusion strategy in the residual decoder. First, we assume that the two input modes have different contributions in the eigenmaps at different scales. Therefore, when fusing multi-modal information, the model will select appropriate features for fusion according to the scale of the feature map and reduce the dimensionality of the multi-modal feature map through the linear layer to extract the most effective information. It is shown in Eq (5):

$$f_{Li} \in \mathbb{R}^{B \times C^i \times H^i \times W^i} = \text{Linear}(f_i^{E1}, f_i^{E2}), \quad i = 1, 2, \dots, 8 \tag{5}$$

where $f_{Li} \in R^{B \times C^i \times H^i \times W^i}$ represents the first $i$ feature map after linear layer processing, and *Linear*($\cdot$) represents the linear layer.

In the second stage of the residual decoder, we use two residual convolutional layers to mine the filtered features further. These residual convolutional layers retain the original feature information while enhancing the fusion effect by learning deeper features. In particular, we use a $3 \times 3$ convolutional layer to finely learn multi-modal features to capture subtle changes in the image. It is shown in Eq (6):

$$f_{Resi} \in \mathbb{R}^{B \times C^i \times H^i \times W^i} = f_{Li} \oplus \text{ResConv}(\text{ResConv}(f_{Li})), \quad i = 1, 2, \dots, 8 \tag{6}$$

where $ResConv(\cdot)$ is the residual convolution and $\oplus$ is the tensor addition. While retaining the $f_{Li}$ information, deeper features are learned through the residual convolutional layer. The specific residual convolutional layer is shown in Eq (7):

$$\text{ResConv}(f_{Li}) = \text{BN}(\text{Conv}(\text{ReLU}(f_{Li}))), \quad i = 1, 2, \dots, 8 \tag{7}$$

Where $f_{Resi} \in R^{B \times C^i \times H^i \times W^i}$ denotes the first $i$ feature map processed by the residual convolutional layer, and it is worth noting that, unlike the downsampled convolutional layer, to learn the multi-modal features more finely, $Conv(\cdot)$ here denotes the $3 \times 3$ convolutional layer.

Finally, in the third stage, we stitch the features of the residual fusion module of the previous layer with the multi-modal features and perform the fusion and upsampling of the feature map through the upsampling layer. This process not only preserves the information of the previous layer but also gradually restores the original size of the image through an upsampling operation. Specifically, it is shown in Eq (8):

$$f_i^D = \text{ReLU}(\text{BN}(\text{ConvT}(\text{Concat}(f_{Resi}, f_{i-1}^D)))), \quad f_i^D \in \mathbb{R}^{B \times C^i \times H^i \times W^i} \tag{8}$$

where $Concat(\cdot)$ represents channel splicing, and $ConvT(\cdot)$ represents $4 \times 4$ transpose convolution operations. With each upsampled convolutional layer, the H and W of the feature map become twice as large. At the same time, the number of channels C gradually decreases from 512 to 1 depending on the depth of the transposed convolutional block.

Finally, after processing with the Tanh activation function, we get the resulting image as shown in Eq (9):

$$\hat{y} = \text{Tanh}(f_8^D) \tag{9}$$

where $\hat{y}$ represents the generated image and $Tanh(\cdot)$ represents the Tanh activation function.

In general, the residual decoder effectively avoids information redundancy and loss problems by fusing the multi-modal information first and then fusing the sorted multi-modal details with the information transmitted from the previous layer. At the same time, introducing residual connections allows the model to retain most of the obtained information while extracting depth features, thereby improving the quality of the generated images.

**Discriminator.** The discriminator part comprises eight convolutional layers, and the overall structure follows the classic PatchGAN architecture [34]. As a mature discriminator framework, PatchGAN is widely popular and widely used in CNN-based generative adversarial networks. To effectively distinguish between modalities, we stitch the generated image output by the generator and the real image of the target modality with the source modal image to form a feature image of $256 \times 256 \times 2$. The feature image is then downsampled and convoluted by five convolution modules, and the feature map size is reduced to 1/4 of the original size after each operation. After processing these convolutional layers, we finally get a one-dimensional $8 \times 8$ feature map, which is presented as a two-dimensional matrix. Each element

of this matrix corresponds to a specific area of the original image. The goal of the discriminator is to make the matrix value of the generated image as close to 0 as possible and the matrix value of the real image as close to 1 as possible to achieve accurate image discrimination. With this design, the discriminator can effectively identify the difference between the generated and real image, thereby assisting the generator to produce a more realistic image.

**Loss function.**   The loss function consists of several components, including generator adversarial loss, discriminator adversarial loss, and pixel reconstruction loss [35]. These loss functions work together to ensure that the generator can produce a high-quality image and that the discriminator can distinguish between the real image and the generated image.

Precisely, generator adversarial loss measures how the image generated by the generator performs in the discriminator. When the discriminator identifies the generated image as a real image (i.e., the output is close to 1), the generator's adversarial loss is minimized. This encourages the generator to produce a more realistic image to trick the discriminator.

$$L_{G\_adv} = \mathbb{E}[D(x, G(x)) - 1] \tag{10}$$

$L_{G\_adv}$ represents the generator adversarial loss, $E(\cdot)$ represents the expectation, $D(\cdot)$ represents the discriminator output, $x$ represents the input image, and $G(x)$ represents the generator output. When the generation result is 1 after being identified by the discriminator, the opponent loss of the generator is minimized.

Discriminator adversarial loss is used to optimize the performance of the discriminator. It contains two parts: one is to ensure that the real image is stitched with the source image and output close to 1 through the discriminator to identify the real image correctly; The other part is to ensure that the generated image is near to 0 after stitching with the source image and output by the discriminator to distinguish the generated image accurately. In this way, the discriminator can continuously improve its discriminatory capabilities.

$$L_{D\_adv} = \mathbb{E}[(D(x, y) - 1)^2] + \mathbb{E}[D(x, G(x))]^2 \tag{11}$$

where $L_{D\_adv}$ represents the discriminator adversarial loss, and $y$ represents the real image.

We have also introduced pixel reconstruction losses to further enhance the realism of the generated image. This loss function evaluates the quality of image reconstruction by calculating the pixel difference between the generated and target real images. By minimizing pixel reconstruction losses, we encourage generators to reconstruct target images more accurately at the pixel level.

$$L_1 = \mathbb{E}[\| y - G(x) \|_1] \tag{12}$$

where $L_1$ represents the pixel reconstruction loss, and $\| \cdot \|_1$ represents the 1 norm.

Ultimately, the overall loss is the weighted sum of these loss functions. By adjusting the pixel reconstruction loss factor and the adversarial loss factor, we can balance the impact of different loss terms on the overall performance.

$$L = \lambda_{L1} L_1 + \lambda_{adv}(L_{G\_adv} + L_{D\_adv}) \tag{13}$$

where $L$ represents the overall loss, $\lambda_{L1}$ represents the pixel reconstruction loss coefficient, and $\lambda_{adv}$ represents the adversarial loss coefficient.

In the training process, the generator and discriminator alternately fix one side to train the other to achieve the purpose of adversarial training. In this way, our method generates high-quality, photorealistic multi-modal images.

We use Peak Signal Noise Ratio (PSNR), Structural Similarity Index (SSIM), and Mutual Information (MI) as evaluation metrics. PSNR quantifies image difference, with higher values indicating better quality. SSIM assesses image structural similarity, and in our research, it specifically indicates the model's prediction accuracy—values closer to 1 mean higher accuracy. MI measures the information overlap between images, with higher values showing stronger correlation.

The author-generated code supporting the findings of this study is available in the GitHub repository at https://github.com/opsliuhao/RIS. The code is released under the MIT License.

## Experiments and results

To verify the quality of the images generated by the proposed multi-modal network, we designed a series of experiments such as comparison experiments with the uni-modal method, ablation experiments, and DTI estimation.

### Comparation

We selected five representative models in the comparative experiments: pix2pix, CycleGAN, pGAN, SwinUnet, and ResViT. We strictly followed each method's experimental parameters, network architecture, and loss function settings in the exposed code. At the same time, all experiments follow the standard of multi-center joint training, and the data set division is completely consistent to ensure that the models can be fairly compared under the same conditions. The experimental environment is an NVIDIA GeForce RTX 3090 graphics card, and the training cycle is unified to 80 epochs.

To comprehensively evaluate each model's performance, we selected peak signal-to-noise ratio, structural similarity, and mutual information as evaluation indicators. These indicators can quantitatively evaluate the images generated by each model in terms of image quality, structure, and information relevance.

Table 4 shows the quantitative results. The method with a high-b-value as the input generally outperforms the method with T1 as the input in the unimodal method. Our RISNet performed excellently on all datasets when using multi-model data as inputs. It achieves the best results, especially on UCDavis, MountSinai, and UWM datasets, which fully proves its advantages in multi-modal image generation tasks.

When using a single-modal image as input for testing, RISNet continues to make predictions and outperforms other traditional models, particularly with high B-value images. This indicates that the model benefits from richer details through joint training with multi-modal data. Fig 2 presents the qualitative results of various methods when high B-value images are used as input.

In summary, our comparative experiments demonstrate the superiority of the proposed multi-modal network in image generation quality. Both quantitative evaluations and qualitative assessments show that our method performs exceptionally well. It effectively leverages the benefits of multi-modal images, resulting in the generation of high-quality low-b-value images.

### Ablation experiments

We carefully designed an ablation experiment to verify the critical role of residual decoders in improving model performance. In this set of experiments, we replaced the residual decoder with a normal upsampled convolutional block to observe its specific impact on the model performance. In the substitution process, we keep the dimensional stitching operation of

**Table 4.** The results of evaluation indicators under different datasets by different methods.

| Data | Methods | UCDavis | | | MountSinai | | | AMU | | | UWM | | |
|---|---|---|---|---|---|---|---|---|---|---|---|---|---|
| | | PSNR | MI | SSIM | PSNR | MI | SSIM | PSNR | MI | SSIM | PSNR | MI | SSIM |
| High-b | Pix2pix | 27.8958 | _1.3882_ | 0.8755 | 26.6304 | **1.3758** | _0.8286_ | 29.8182 | 1.3332 | _0.9179_ | _37.0618_ | 1.4395 | 0.9666 |
| | CycleGAN | 29.1510 | **1.4230** | 0.8802 | 24.3045 | 1.3371 | 0.7845 | **30.7778** | **1.3434** | **0.9297** | 36.4730 | 1.4177 | 0.9574 |
| | Pgan | 26.3843 | 1.3684 | 0.8657 | _28.3447_ | 1.3172 | 0.7903 | 25.9253 | 1.3250 | 0.8916 | 36.4890 | _1.4435_ | 0.9581 |
| | SwinUnet | _30.9245_ | 1.3822 | _0.8923_ | 26.4684 | 1.3116 | 0.8032 | 27.0575 | 1.3246 | 0.8854 | 36.8960 | 1.4410 | _0.9670_ |
| | ResViT | 26.1521 | 1.3498 | 0.8744 | 27.9977 | 1.3018 | 0.7765 | 25.8866 | _1.3175_ | 0.9005 | 30.3188 | 1.3734 | 0.9333 |
| | RISNet | **31.4683** | 1.3859 | **0.8999** | **29.1133** | _1.3412_ | **0.8496** | 29.1261 | 1.3337 | 0.9101 | **37.9960** | **1.4601** | **0.9699** |
| T1 | Pix2pix | 28.3444 | 1.3548 | 0.8740 | **28.5952** | **1.3220** | _0.8093_ | _28.0340_ | 1.3146 | 0.8994 | 34.5537 | 1.4057 | 0.9553 |
| | CycleGAN | 17.6904 | 1.3274 | 0.8346 | 22.1912 | 1.2539 | 0.6807 | 19.5261 | 1.3033 | 0.8640 | 23.1217 | 1.3643 | 0.9143 |
| | Pgan | 20.4207 | 1.3190 | 0.8415 | 19.5355 | 1.2434 | 0.6326 | 22.6663 | 1.2997 | 0.8705 | 19.5859 | 1.3262 | 0.9106 |
| | SwinUnet | 28.2720 | 1.3526 | 0.8683 | 26.0110 | 1.3074 | 0.7770 | **29.4108** | _1.3174_ | **0.9026** | 32.6220 | 1.3878 | 0.9490 |
| | ResViT | **30.6301** | **1.3724** | _0.8958_ | 26.1786 | 1.2615 | 0.7339 | 23.4336 | **1.3253** | 0.8623 | _34.9804_ | _1.4126_ | _0.9572_ |
| | RISNet | 29.2605 | _1.3682_ | **0.8991** | 28.1000 | _1.3202_ | **0.8460** | 27.1652 | 1.3005 | _0.9003_ | **38.1688** | **1.4506** | **0.9642** |
| Both | RISNet | 32.0891 | 1.3982 | 0.9039 | 29.1681 | 1.3611 | 0.8558 | 28.4900 | 1.3208 | 0.9146 | 39.5037 | 1.4611 | 0.9729 |

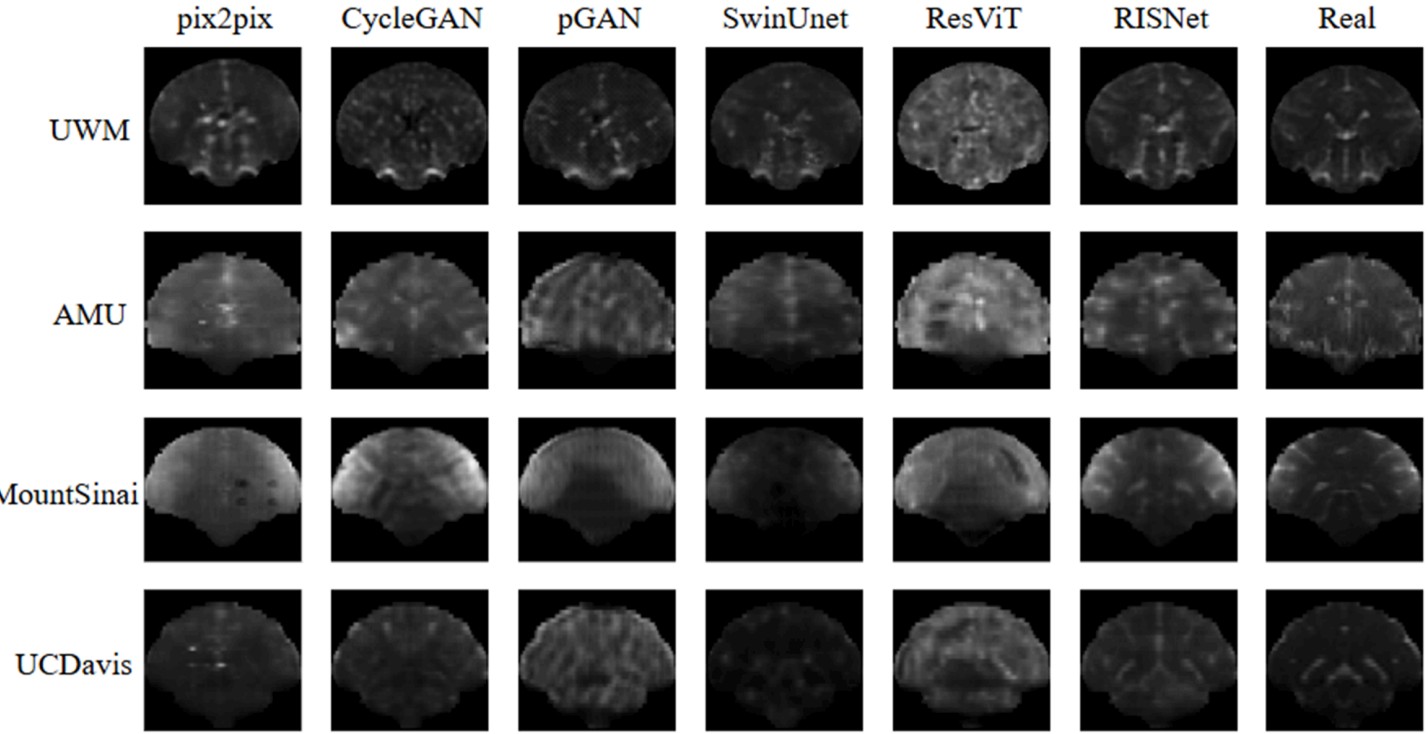

**Fig 2. Compare experiments.**

the multi-modal feature map and the sampling convolutional block on the previous layer unchanged to ensure the consistency of experimental conditions. The design details of this decoder are shown in Eq (14):

$$f_i^D = BN(Conv(ReLU(Concat(f_i^{E1}, f_i^{E2}, f_{i-1}^D))))), \quad i = 1, 2, \dots, 8 \tag{14}$$

The experimental results are shown in Fig 3 and Table 5. After replacing the residual decoder on the UWM dataset, the number and quality of the generated image texture features

are significantly reduced. This is directly reflected in the decline of various evaluation indicators, which fully proves the effectiveness of the residual decoder in improving the texture detail of the image.

Although both methods can only roughly learn texture details on the AMU dataset, the image generated by the model using the residual decoder is more similar in color to the reference image. However, in terms of evaluation indicators, the results after removing the residual decoder are improved, possibly due to the quality and quantity of the AMU dataset itself and the low similarity with other site data. However, this does not mean that replacing the residual decoder is a better option, as the residual decoder has better generalization capabilities.

After replacing the residual decoder, the model performs poorly on the MountSinai and UCDavis datasets. On the MountSinai dataset, the model failed to effectively learn the texture information, resulting in a significant decline in various indicators. On the UCDavis dataset, the model even learns unreal texture information after replacing the residual decoder, which further widens the gap with the reference image and reduces the evaluation index.

In summary, this set of ablation experiments has identified the residual decoder's crucial role in enhancing the model's texture detail and generalization ability.

## DTI estimation experiments

DTI imaging estimates the diffusion tensor by applying gradient pulses in different directions to measure water molecules' diffusion in all directions. Based on the measured

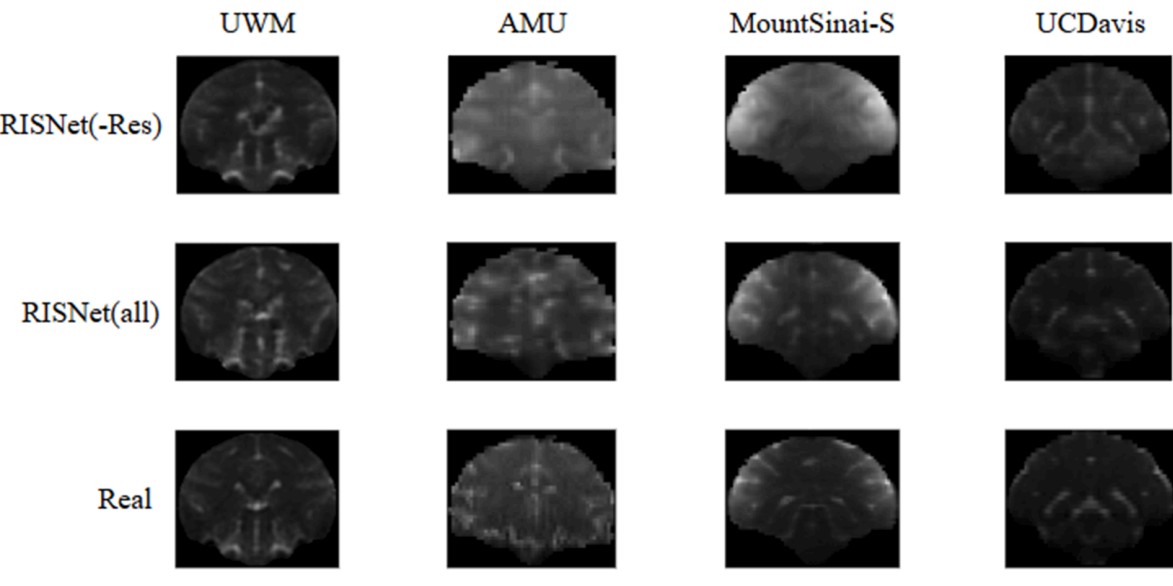

**Fig 3. Ablation experiments.**

**Table 5. Ablation experiments.**

| Methods | UCDavis | | | MountSinai | | | AMU | | | UWM | | |
|---|---|---|---|---|---|---|---|---|---|---|---|---|
| | PSNR | MI | SSIM | PSNR | MI | SSIM | PSNR | MI | SSIM | PSNR | MI | SSIM |
| RISNet(-Res) | 25.8444 | 1.3852 | 0.8690 | 24.6560 | 1.3546 | 0.8145 | 31.5442 | 1.3354 | 0.9257 | 37.9029 | 1.4494 | 0.9695 |
| RISNet(all) | 32.0891 | 1.3982 | 0.9039 | 29.1681 | 1.3611 | 0.8558 | 28.4899 | 1.3208 | 0.9146 | 39.5037 | 1.4611 | 0.9729 |

diffusion-weighted imaging data, the diffusion tensor parameters for each voxel can be calculated using a linear regression method, such as the principal diffusion direction, the eigenvalues of the diffusion tensor, and the eigenvectors. The diffusion parameter DTI provides a variety of diffusion parameters, the most commonly used of which is Fractional Anisotropy (FA), which indicates the degree of directionality of the diffusion of water molecules in the tissue. The experimental results are shown in Fig 4. The texture of the FA images estimated from each dMRI dataset is clear, but they contain a significant amount of noise. By incorporating low-b-value images generated through various methods, the noise levels in the FA images are reduced to varying extents. Notably, our RISNet approach effectively removes noise while better preserving the sharp and highlighted texture details of the original FA image. In contrast, although the SwinUnet method demonstrates some denoising capabilities, it inadvertently introduces additional unwanted noise.

To further validate the quality of the low-b-value images generated by our method, we performed in-depth DTI estimation experiments. In the experiment, we fused the low-b-value images generated by different methods into the dMRI images and added a low-b-value volume to the dMRI images. On the UCDavis dataset, we obtained the dMRI images enhanced by different methods. We estimated the DTI of these images, while on the MountSinai-S dataset, we replaced the original low-b-value images in MountSinai-S with the low-b-value images generated by the ResViT and RISNet for Xtract tracking experiments. The experimental results are shown in Fig 5.

In the Xtract tracking experiment, we found that replacing the Xtract outputs with low-b-value images generated by our proposed method produced results that closely resemble those of the original diffusion MRI (dMRI) data. In contrast, the results from the ResViT method deviated significantly from the original dMRI images.

Through experiments involving DTI estimation, we confirmed the advantages of our RISNet method in generating high-quality low-b-value images. Our approach enhances the visual quality of FA images and more accurately reflects the characteristics of the tissue's microstructure.

## Discussion

In this study, we innovatively propose RISNet for macaque dMRI image conversion and data enhancement. The RIS encoder part is extensible, and a new encoder path with the same encoder path can be added to have one more modal input. In addition, this approach's core lies in its residual decoder's design, which enables the effective fusion of information in three steps. Firstly, the decoder fuses the multi-modal feature map from the encoder and

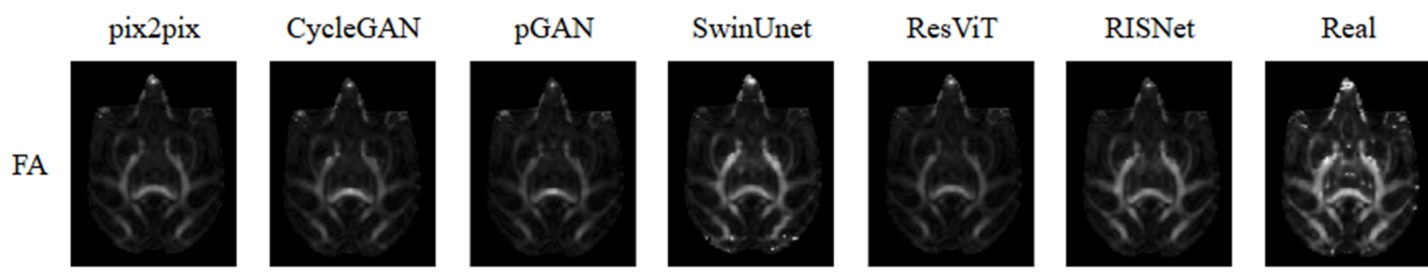

**Fig 4. DTI estimation experiments.**

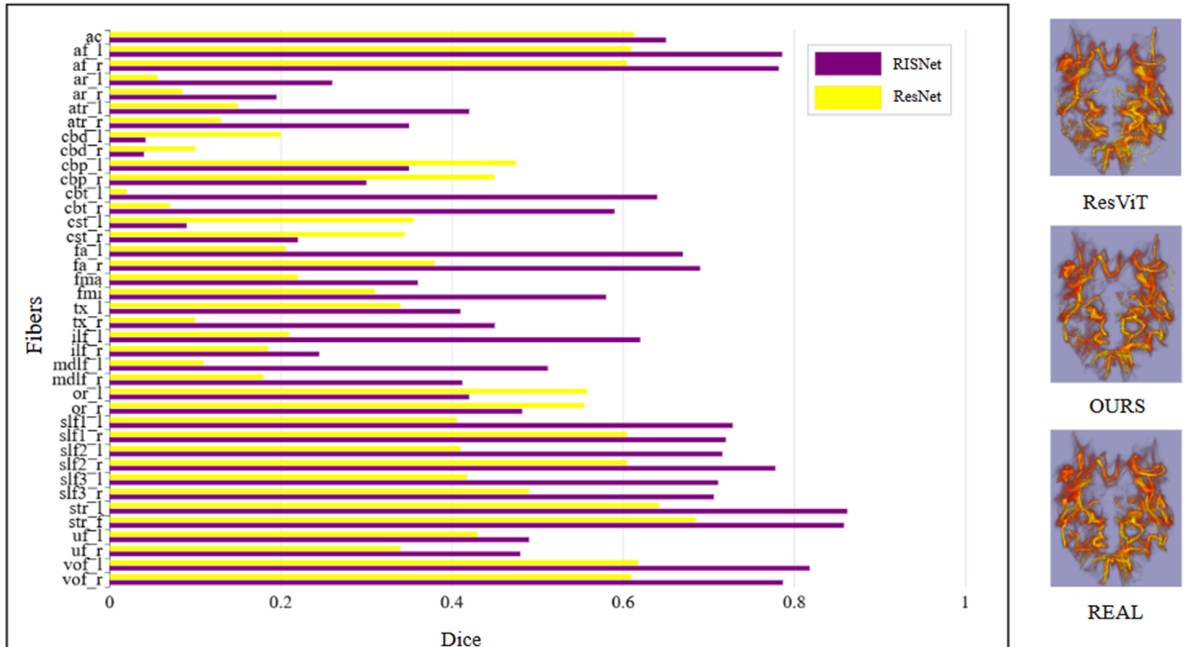

**Fig 5. Results of Xtract calculations.**

reduces the dimensionality of the multi-modal features. Subsequently, the residual convolutional structure is used to learn more and deeper information while retaining the existing multi-modal information. Finally, the decoder further fuses the fused multi-modal information with the previous layer's feature map to filter out the key information with more details among the many redundant information.

The greatest advantage of RIS is that it does not interfere with the pre-trained model, and all features will be processed into an average feature vector before the network learning. Therefore, when multi-modal data is added, there is no need to retrain the pre-trained model. Thus, data of other modalities can be added for training at any time.

Considering that the sample size of macaque data at each site is relatively small, but the data from each site show intraspecific consistency, we adopted a multi-center joint training method to alleviate the problem caused by sample size. However, the differences in acquisition parameters between different sites put forward high requirements for the model's generalization ability. Experimental results show that the traditional uni-modal method performs poorly in image generation details. Specifically, Pix2pix retains multi-scale information to a certain extent through the U-Net network structure. CycleGAN's cyclic consistency loss performs well in unsupervised scenarios. Although pGAN can learn information about deep networks by using ResNet as a generator, it ignores the advantages of U-Net in retaining multi-scale information. SwinUnet and ResViT capture the contextual relationship between pixels through an attention mechanism. However, as shown in the figure, due to the limitations of single-modal data, these methods are relatively general in performance, and their generalization ability is weak under the condition of multi-center joint training.

In contrast, RISNet is based on the U-Net architecture, introducing two encoders to read data in two modalities. The residual decoder fuses multi-scale and multi-modal information

hierarchically, generating an image that performs better in detail texture. At the same time, it can still maintain good generalization ability in joint training.

Ablation experiments further demonstrated the effectiveness of the residual decoder. When we replaced the multi-modal residual fusion module with simple dimension stitching and upsampling convolution blocks, both the qualitative and quantitative results of the model significantly declined, and its generalization ability was greatly diminished. This indicates that the basic upsampled convolutional block is not effective in extracting important features from the multi-modal feature map, nor does it manage the redundant information present in the previous convolutional blocks. In contrast, our designed residual decoder substantially enhances the efficiency of multi-modal fusion by first reducing the dimensionality of the multi-modal features to extract relevant information. It then fuses these reduced features with those extracted from the convolutional blocks of the previous layer, facilitating the extraction of more detailed information.

In addition, the experimental results of DTI estimation indirectly prove that the quality of the low-b-value images generated by our method is high, which is of positive significance for subsequent analyses such as nerve fiber tract tracing. It is worth noting that, unlike the images mapped to the RGB color space, the evaluation of 3D medical images should not be limited to visual observation and the calculation of quantitative evaluation indicators, and further analysis of the generated images is also an effective evaluation method [36].

As we move forward, we plan to broaden our approach to human brain imaging. While most existing human brain imaging data do not typically suffer from an imbalance between low-b-value and high-b-value images, this area remains crucial for medical research. Presently, most methods for converting human brain images rely on single-modal images. Although some progress has been made, these methods do not fully leverage the benefits of multi-modal data. Therefore, future research in multi-modal human brain image conversion will be essential. We aim to achieve more innovations and breakthroughs in medical image analysis by continuously refining and enhancing our methodology.

## Conclusion

In this paper, we propose a variable multi-modal image feature fusion adversarial neural network called RISNet, which uses rapid insertion structural to input features from different modes into a general residual decoding structure for multi-modal macaque brain image conversion and data enhancement. In the encoder part of the generator, we downsample and extract multi-scale feature maps from T1 and high-b-value images through two identical convolutional downsampling paths. In particular, our method is not limited to the input of two modalities, and a new encoder path can receive one more modal image. In the decoder part of the generator, we first fuse and reduce the multi-modal information through the three-step fusion method, then further mine the multi-modal information through the residual network, and finally fuse the information obtained by the previous layer of decoding and multi-modal information, to eliminate redundant information and learn more image detail information. Finally, we verified the method's effectiveness through comparative, ablation, and DTI estimation experiments. Our method can be applied to generate low-b-value images of macaques and enhance dMRI image data to provide more high-quality dMRI images for brain science research.

## Acknowledgments

We would like to express our gratitude to Professor Li Haifang from Taiyuan University of Technology for providing financial support for this research and offering meticulous guidance and rigorous review during the research design, experiment implementation, and paper writing processes. Professor Li's professional insights and rigorous attitude have significantly improved the quality of this study, and we hereby extend our sincere thanks.

## Author contributions

**Data curation:** Xiaohong Xue.

**Methodology:** Yifei Chen.

**Supervision:** Haifang Li.

**Visualization:** Hao Liu.

**Writing – original draft:** Guolan Wang.

**Writing – review & editing:** Qianshan Wang.

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
