## [Decision Letter · Decision Letter 0]

8 Apr 2025

PONE-D-25-02963RISNet: A Variable Multi-modal Image Feature Fusion Adversarial Neural Network for Generating Specific dMRI ImagesPLOS ONE

Dear Dr. hao,

Thank you for submitting your manuscript to PLOS ONE. After careful consideration, we feel that it has merit but does not fully meet PLOS ONE’s publication criteria as it currently stands. Therefore, we invite you to submit a revised version of the manuscript that addresses the points raised during the review process.

We look forward to receiving your revised manuscript.

Kind regards,

Vatsala Anand

Academic Editor

PLOS ONE

 [This work was supported by the Science and Technology Innovation Project of Shanxi Provincial Colleges and Universities (2023L494), the National Natural Science Foundation of China (61976150), and Shanxi Provincial Natural Science Foundation Youth Fund Project(2021030212427).]. 

[This work was supported by the Science and Technology Innovation Project of Shanxi Provincial Colleges and Universities (2023L494), the National Natural Science Foundation of China (61976150), and Shanxi Provincial Natural Science Foundation Youth Fund Project(2021030212427).]

 [This work was supported by the Science and Technology Innovation Project of Shanxi Provincial Colleges and Universities (2023L494), the National Natural Science Foundation of China (61976150), and Shanxi Provincial Natural Science Foundation Youth Fund Project(2021030212427).]

5.  Please amend the manuscript submission data (via Edit Submission) to include author Qianshan

Wang.

6.  We notice that your supplementary figures are uploaded with the file type 'Figure'. Please amend the file type to 'Supporting Information'. Please ensure that each Supporting Information file has a legend listed in the manuscript after the references list.

Additional Editor Comments:

Reviewer 1:

Following issues have existed into submitted article as follows:

1-The contribution of the presented paper is not clear. Please revise section 1 and add your valuable innovation there.

2-The methodology should be organized where an introduction for this section should be included as well as a main diagram that represents the whole steps also should be provided.

3-Term "Participants" into table 1 refers for what?if refers for human involvement into data acquisition where is the authorization letter?

4-Authors mentioned 'We selected five representative models in the comparative experiments: pix2pix, 288

CycleGAN, pGAN, SwinUnet, and ResViT".My question why they chose these methods as baseline for comparisons. How far these methods near to proposed study and extracted from previous works. Please cite all selected references after justify previous concerns.

5-The evaluation part focused only one the image enhancement part and ignore model performance for instance we do not know the achieved test accuracy or if there is unbalance data issue that resulted in misleading results. In other word, more focus is needed on deep learning evaluation perspective.

Reviewer 2:

The manuscript is well organized with good English language. The proposed approach is promising with a significant advance compared to the existing approaches. The study covered the related existing sources. The manuscript can be accepted for publication in its current form.

Academic editor:

1. How does the imbalance between lower and higher b-value images specifically impact the accuracy of computational neuroscience studies?

2. What challenges arise from the multi-center and small-sample problems in the macaque dMRI brain dataset, and how do they affect model training?

3. How does RISNet improve upon existing generative adversarial network (GAN)-based medical image conversion methods?

4. What are the specific advantages of the rapid insertion structural (RIS) mechanism in multi-modal feature fusion?

5. Why were T1 images and higher b-value images chosen as inputs for generating lower b-value images, and how does this choice impact model performance?

6. What are the underlying reasons behind the observed improvements in PSNR and SSIM indices using RISNet compared to other methods?

7. How does the proposed method contribute to the broader field of neuroscience research beyond macaque dMRI image conversion?

8. Were there any limitations or constraints in the study that could affect the reproducibility or generalization of the model to other datasets?

Reviewers' comments:

Reviewer's Responses to Questions

**Comments to the Author**

1. Is the manuscript technically sound, and do the data support the conclusions?

Reviewer #1: Yes

Reviewer #2: Yes

2. Has the statistical analysis been performed appropriately and rigorously? 

Reviewer #1: Yes

Reviewer #2: Yes

3. Have the authors made all data underlying the findings in their manuscript fully available?

Reviewer #1: Yes

Reviewer #2: Yes

4. Is the manuscript presented in an intelligible fashion and written in standard English?

Reviewer #1: Yes

Reviewer #2: Yes

5. Review Comments to the Author

Reviewer #1: The manuscript is well organized with good English language. The proposed approach is promising with a significant advance compared to the existing approaches. The study covered the related existing sources. The manuscript can be accepted for publication in its current form.

Reviewer #2: Following issues have existed into submitted article as follows:

1-The contribution of the presented paper is not clear. Please revise section 1 and add your valuable innovation there.

2-The methodology should be organized where an introduction for this section should be included as well as a main diagram that represents the whole steps also should be provided.

3-Term "Participants" into table 1 refers for what?if refers for human involvement into data acquisition where is the authorization letter?

4-Authors mentioned 'We selected five representative models in the comparative experiments: pix2pix, 288

CycleGAN, pGAN, SwinUnet, and ResViT".My question why they chose these methods as baseline for comparisons. How far these methods near to proposed study and extracted from previous works. Please cite all selected references after justify previous concerns.

5-The evaluation part focused only one the image enhancement part and ignore model performance for instance we do not know the achieved test accuracy or if there is unbalance data issue that resulted in misleading results. In other word, more focus is needed on deep learning evaluation perspective.

6. PLOS authors have the option to publish the peer review history of their article (what does this mean?). If published, this will include your full peer review and any attached files.

Reviewer #1: **Yes: **Ghassan Abdul-Majeed

Reviewer #2: No

---

## [Author Response · Author response to Decision Letter 1]

15 May 2025

Answer:Thank you for the reminder. I've carefully revised my manuscript to meet all of PLOS ONE's style requirements, including those for file naming. I'm looking forward to the next steps in the submission process.

2.Please note that PLOS ONE has specific guidelines on code sharing for submissions in which author-generated code underpins the findings in the manuscript. In these cases, we expect all author-generated code to be made available without restrictions upon publication of the work. Please review our guidelines at https://journals.plos.org/plosone/s/materials-and-software-sharing#loc-sharing-code and ensure that your code is shared in a way that follows best practice and facilitates reproducibility and reuse.

Answer The author-generated code supporting the findings of this study is available in the GitHub repository at https://github.com/opsliuhao/RIS. The code is released under the MIT License

 [This work was supported by the Science and Technology Innovation Project of Shanxi Provincial Colleges and Universities (2023L494), the National Natural Science Foundation of China (61976150), and Shanxi Provincial Natural Science Foundation Youth Fund Project(2021030212427).]. 

Answer We have added the funding information in the cover letter.

[This work was supported by the Science and Technology Innovation Project of Shanxi Provincial Colleges and Universities (2023L494), the National Natural Science Foundation of China (61976150), and Shanxi Provincial Natural Science Foundation Youth Fund Project(2021030212427).]

 [This work was supported by the Science and Technology Innovation Project of Shanxi Provincial Colleges and Universities (2023L494), the National Natural Science Foundation of China (61976150), and Shanxi Provincial Natural Science Foundation Youth Fund Project(2021030212427).]

Answer We have removed the funding information from the "Acknowledgments" section in the article.

5. Please amend the manuscript submission data (via Edit Submission) to include author Qianshan Wang.

Answer We have added the author Qianshan Wang to the "Authors" section. He/she appears as the corresponding author in the sixth position. Please check.

6.  We notice that your supplementary figures are uploaded with the file type 'Figure'. Please amend the file type to 'Supporting Information'. Please ensure that each Supporting Information file has a legend listed in the manuscript after the references list.

Answer: We have modified the file type of the online - submitted images to "Supporting Information" as required. Additionally, we have provided a corresponding legend for each supplementary information file after the references in the article.

Reviewer 1:

Following issues have existed into submitted article as follows:

1-The contribution of the presented paper is not clear. Please revise section 1 and add your valuable innovation there.

Answer:This paper introduces a Rapid Insertion Structure (RIS) combined with residual decoders to the issues above by integrating the benefits of multi-modal data using the GAN method. The novel residual decoder employed in this paper exhibits robust feature extraction and expression capabilities, facilitating precise conversion from high-b-value images and T1 images to low-b-value images. This approach enables the generation of high-quality images to compensate for the limited quantity of low-b-value images in dMRI data, leveraging the full potential of multi-modal medical imaging data to enhance the precision and dependability of macaque brain image analysis. The key contributions of this study are outlined as follows:

1. Innovative Architecture for Multi-Modal Integration: we innovatively proposed a rapid insertion structure that requires only simple modifications to the model to extract and integrate various modal data. This mode can be rewritten effortlessly to combine features from various data types. This approach enables the application of multi-center joint training in expanding the macaque dMRI dataset during the experiment.

2. Residual Decoder-Based Multi-Modal Fusion: A novel residual decoder framework is proposed for multi-modal dMRI image conversion. The residual decoder carries out multi-modal fusion and decoder feature fusion, which avoids the loss of information caused by information redundancy, makes full use of multi-modal information, and generates a more accurate diffusion MRI.

2-The methodology should be organized where an introduction for this section should be included as well as a main diagram that represents the whole steps also should be provided.

Answer: We have added an overall flowchart to the article.

3-Term "Participants" into table 1 refers for what?if refers for human involvement into data acquisition where is the authorization letter?

Answer: The terms in the article have been revised. The datasets used in this article are from different sites, and there are significant differences in the number of samples for each dataset.

4-Authors mentioned 'We selected five representative models in the comparative experiments: pix2pix, 288

CycleGAN, pGAN, SwinUnet, and ResViT".My question why they chose these methods as baseline for comparisons. How far these methods near to proposed study and extracted from previous works. Please cite all selected references after justify previous concerns.

Answer: We chose pix2pix, CycleGAN, pGAN, SwinUnet, and ResViT as baseline models for the comparative experiments because they are highly regarded and have demonstrated excellent performance in various aspects of image generation and transformation in recent years. These models represent different approaches and techniques within the field, covering both generative adversarial network - based methods and architectures inspired by transformers. By comparing our proposed method with these well - established models, we can comprehensively evaluate its effectiveness and competitiveness across a wide range of relevant tasks and techniques. This allows us to provide a more robust and reliable assessment of our model's performance in the context of current state - of - the - art research in image generation and transformation.

5-The evaluation part focused only one the image enhancement part and ignore model performance for instance we do not know the achieved test accuracy or if there is unbalance data issue that resulted in misleading results. In other word, more focus is needed on deep learning evaluation perspective.

Answer: We use Peak Signal Noise Ratio (PSNR), Structural Similarity Index (SSIM), and Mutual Information (MI) as evaluation metrics. PSNR quantifies image difference, with higher values indicating better quality. SSIM assesses image structural similarity, and in our research, it specifically indicates the model's prediction accuracy—values closer to 1 mean higher accuracy. MI measures the information overlap between images, with higher values showing stronger correlation.

Reviewer 2:

The manuscript is well organized with good English language. The proposed approach is promising with a significant advance compared to the existing approaches. The study covered the related existing sources. The manuscript can be accepted for publication in its current form.

Academic editor:

1. How does the imbalance between lower and higher b-value images specifically impact the accuracy of computational neuroscience studies?

Answer:In some DTI calculation methods, calibration with low b-value images is performed every 12 to 15 volumes. To accommodate this, certain DWI acquisition sequences add extra low b-value images proportionally based on the total number of volumes acquired. Others automatically acquire low b-value images at intervals of 12 to 15 volumes. These configurations are specifically designed to meet the requirements of DTI calculations. Failure to satisfy this condition may lead to inaccurate DTI numerical results.

2. What challenges arise from the multi-center and small-sample problems in the macaque dMRI brain dataset, and how do they affect model training?

Answer:Medical images are not recorded in grayscale color spaces; instead, all medical imaging data records signal intensity values or other computationally derived numerical values. Due to significant variations in raw data distributions caused by different acquisition protocols, equipment, or even daily environmental conditions during scanning, MRI signal distributions exhibit substantial discrepancies across multi-center datasets. This poses significant challenges for deep neural networks relying on statistical learning, as models require larger datasets to analyze the data distribution characteristics of target regions. Additionally, rhesus macaque datasets often suffer from small-sample limitations, making it difficult to ensure sufficient data volume during training with conventional models.

3. How does RISNet improve upon existing generative adversarial network (GAN)-based medical image conversion methods?

Answer:In traditional feature channel fusion methods, different modal data are generally subjected to the same feature extraction pattern, which fails to fully leverage the diverse characteristics of multimodal data. To address this, RISNet employs its RIS (Reconfigurable Intelligent Surface) architecture to extract features from signal data of different modalities. Specifically, data from distinct modalities undergo feature extraction through corresponding branches in the RIS structure, and features obtained via different extraction methods are fused. This approach accelerates model convergence and ensures good generalization performance.Meanwhile, RISNet incorporates the concept of residual structures but adjusts their scope of action, confining the propagation of captured residual modules within their respective RIS branches. This strategy mitigates the risk of gradient explosion caused by the expanded solution space due to multimodal data, effectively enhancing the model's stability and training efficiency.

4. What are the specific advantages of the rapid insertion structural (RIS) mechanism in multi-modal feature fusion?

Answer:The RIS (Reconfigurable Intelligent Surface) structure extracts features from signal data of different modalities through corresponding branch pathways. Specifically, data from distinct modalities undergo feature extraction in their dedicated RIS branches, and features derived from different extraction methods are fused. This design accelerates model convergence and ensures strong generalization performance.Additionally, RISNet incorporates the concept of residual structures but adjusts their operational scope, confining the propagation of captured residual modules within their respective RIS branches. This adjustment mitigates the risk of gradient explosion caused by the expanded solution space due to multimodal data, effectively enhancing the model's training stability and efficiency.

5. Why were T1 images and higher b-value images chosen as inputs for generating lower b-value images, and how does this choice impact model performance?

Answer:RISNet can accept not only T1 and high b-value images as inputs but also use T2 in correspondence with T1. By adding RIS branches, it enables simultaneous input of T1, T2, and high b-value images, which stems from the excellent characteristics of the RIS module. In this study, T1 and high b-value data were adopted because the vast majority of the datasets collected during the research process only included T1 and high b-value images for training. Using other modal data showed no significant impact on the training process.

6. What are the underlying reasons behind the observed improvements in PSNR and SSIM indices using RISNet compared to other methods?

Answer:Compared to other models, the fundamental reason for the performance improvement of this method lies in the ability of different RIS branches to apply distinct feature processing weights to data from different modalities, thereby enhancing model performance. However, this approach may expand the solution space of the target problem. Therefore, we introduce the concept of residual structures to minimize the occurrence of this phenomenon.

7. How does the proposed method contribute to the broader field of neuroscience research beyond macaque dMRI image conversion?

Answer:We have preliminarily applied this method to DWI data of partial human brains and found that it can quickly achieve model performance with similar accuracy on human brain data. This provides a viable tool for the restoration and quality improvement of human brain data.

8. Were there any limitations or constraints in the study that could affect the reproducibility or generalization of the model to other datasets?

Answer:In this study, no restrictive conditions were imposed on the image data itself; all improvement methods were derived from analyses of the characteristics of medical images. However, our research and experiments were limited to multiple rhesus macaque datasets provided by PRIME-DE, and we are unaware of whether these datasets underwent significant preprocessing. Therefore, whether the RISNet proposed in this paper can achieve similar performance on other datasets remains to be further investigated.

---

## [Decision Letter · Decision Letter 1]

26 May 2025

PONE-D-25-02963R1RISNet: A Variable Multi-modal Image Feature Fusion Adversarial Neural Network for Generating Specific dMRI ImagesPLOS ONE

Dear Dr. hao,

Thank you for submitting your manuscript to PLOS ONE. After careful consideration, we feel that it has merit but does not fully meet PLOS ONE’s publication criteria as it currently stands. Therefore, we invite you to submit a revised version of the manuscript that addresses the points raised during the review process.

As attached below, there are still some minor issues that have haven't been fully addressed yet. Please follow the suggestions to your best efforts.

We look forward to receiving your revised manuscript.

Kind regards,

Wang Zhan, Ph.D.

Academic Editor

PLOS ONE

Journal Requirements:

Reviewers' comments:

Reviewer's Responses to Questions

**Comments to the Author**

1. If the authors have adequately addressed your comments raised in a previous round of review and you feel that this manuscript is now acceptable for publication, you may indicate that here to bypass the “Comments to the Author” section, enter your conflict of interest statement in the “Confidential to Editor” section, and submit your "Accept" recommendation.

Reviewer #1: All comments have been addressed

Reviewer #2: (No Response)

2. Is the manuscript technically sound, and do the data support the conclusions?

Reviewer #1: Yes

Reviewer #2: (No Response)

3. Has the statistical analysis been performed appropriately and rigorously? 

Reviewer #1: Yes

Reviewer #2: (No Response)

4. Have the authors made all data underlying the findings in their manuscript fully available?

Reviewer #1: Yes

Reviewer #2: (No Response)

5. Is the manuscript presented in an intelligible fashion and written in standard English?

Reviewer #1: Yes

Reviewer #2: (No Response)

6. Review Comments to the Author

Reviewer #1: The authors addressed the reviewer comments, and the revised copy is much better than the original one. I think it deserves the acceptance of publication.

Reviewer #2: Still some comments not addressed well such as:

2-The methodology should be organized where an introduction for this section should be included as well as a main diagram that represents the whole steps also should be provided. Authors have add a very basic Flowchart however, this flowchart still need to modify by add each phase of proposed methodology then each sub-process.

4-Authors mentioned 'We selected five representative models in the comparative experiments: pix2pix, 288

CycleGAN, pGAN, SwinUnet, and ResViT".My question why they chose these methods as baseline for comparisons. How far these methods near to proposed study and extracted from previous works. Please cite all selected references after justify previous concerns. The justification is not enough since there is no cited works. However, the baseline works should be chose without any bias that show proposed study is the best. At least authors should be compare their work with previous model on the same case study. therefore

5-The evaluation part focused only one the image enhancement part and ignore model performance for instance we do not know the achieved test accuracy or if there is unbalance data issue that resulted in misleading results. In other word, more focus is needed on deep learning evaluation perspective. The authors response still not convincing.

7. PLOS authors have the option to publish the peer review history of their article (what does this mean?). If published, this will include your full peer review and any attached files.

Reviewer #1: **Yes: **Ghassan Abdul-Majeed

Reviewer #2: No

---

## [Author Response · Author response to Decision Letter 2]

27 Jun 2025

Reviewer #2: 

2-The methodology should be organized where an introduction for this section should be included as well as a main diagram that represents the whole steps also should be provided. Authors have add a very basic Flowchart however, this flowchart still need to modify by add each phase of proposed methodology then each sub-process.

Answer: We sincerely apologize for the previously oversimplified diagrams. We have now redrawn and refined them to comprehensively cover the overall workflow while detailing the specifics of each module. Additionally, we have supplemented the text with a dedicated introduction to the model training process.

Specifically:

Figure A illustrates the data flow across all modules.

Figure B focuses on the generator, with explicit annotation of the RIS structure.

Figure C depicts the discriminator following the PatchGAN architecture.

Figure D provides an in-depth view of the residual decoder structure within RIS.

4-Authors mentioned 'We selected five representative models in the comparative experiments: pix2pix, CycleGAN, pGAN, SwinUnet, and ResViT".My question why they chose these methods as baseline for comparisons. How far these methods near to proposed study and extracted from previous works. Please cite all selected references after justify previous concerns. The justification is not enough since there is no cited works. However, the baseline works should be chose without any bias that show proposed study is the best. At least authors should be compare their work with previous model on the same case study. Therefore

Answer: Pix2Pix and CycleGAN, as foundational models in the field of GANs, achieved breakthroughs in supervised and unsupervised image-to-image translation tasks. Pix2Pix, by introducing local consistency constraints through conditional GANs, was applied by Sun et al. (2022) to denoise low-dose myocardial perfusion SPECT images, significantly improving metrics such as structural similarity and noise coefficients, thus enhancing diagnostic performance[1]. CycleGAN, on the other hand, utilized cycle-consistency loss to enable image translation without paired data. The POCS-Augmented CycleGAN method combined CycleGAN with the traditional Projection Onto Convex Sets (POCS) algorithm, demonstrating superior performance in reconstructing undersampled MR images compared to classical approaches[2].pGAN (Progressive GAN) advanced generative models by employing progressively growing network layers to generate high-resolution images. The multi-stage P-GAN proposed by Mahapatra and Bozorgtabar (2019) significantly improved the detection accuracy of anatomical landmarks and lesions in retinal vasculature and cardiac MRI images through the use of triplet loss[3].

With the rise of Transformer architectures, SwinUNet combined U-Net with Swin Transformer to propose a pure Transformer-based U-shaped structure. This model demonstrated outstanding performance in multi-organ and cardiac CT image segmentation tasks by effectively learning both local and global semantic features[4].

Finally, ResViT integrated residual learning with Transformer architectures, combining the precision of convolutional networks with the global perceptual capabilities of Transformers[5]. It excelled in cross-modal medical image synthesis tasks, such as MRI-to-CT translation. By employing an aggregated residual Transformer module as the bottleneck structure, the model surpassed traditional methods in both quantitative and qualitative synthesis metrics.

The advantage of Transformers lies in their superior ability to correlate features across different spatial positions in an image, but they have higher memory requirements. CNNs, on the other hand, process images faster but are weaker in capturing spatial associations. This study combines CNNs to enhance their spatial association capabilities for medical image translation. Therefore, we selected Transformer-based models and some adversarial generative network models as baseline models for comparison.

[1] Sun, J., Du, Y., Li, C., Wu, T. H., Yang, B., & Mok, G. S. P. (2022). Pix2Pix generative adversarial network for low dose myocardial perfusion SPECT denoising. Quantitative imaging in medicine and surgery, 12(7), 3539–3555.

[2] Li, Y., Yang, H., Xie, D., Dreizin, D., Zhou, F., & Wang, Z. (2022). POCS-Augmented CycleGAN for MR Image Reconstruction. Applied sciences (Basel, Switzerland), 12(1), 114.

[3] Mahapatra, D., & Bozorgtabar, B. (2019). Progressive Generative Adversarial Networks for Medical Image Super resolution. ArXiv, abs/1902.02144.

[4] Hu Cao, Yueyue Wang, Joy Chen, Dongsheng Jiang, Xiaopeng Zhang, Qi Tian, and Manning Wang. 2022. Swin-Unet: Unet-Like Pure Transformer for Medical Image Segmentation. In Computer Vision – ECCV 2022 Workshops: Tel Aviv, Israel, October 23–27, 2022, Proceedings, Part III. Springer-Verlag, Berlin, Heidelberg, 205–218.

[5] Dalmaz, O., Yurt, M., & Cukur, T. (2022). ResViT: Residual Vision Transformers for Multimodal Medical Image Synthesis. IEEE transactions on medical imaging, 41(10), 2598–2614.

5-The evaluation part focused only one the image enhancement part and ignore model performance for instance we do not know the achieved test accuracy or if there is unbalance data issue that resulted in misleading results. In other word, more focus is needed on deep learning evaluation perspective. The authors response still not convincing.

Answer:

This study focuses on medical image modality translation rather than image enhancement. Common deep learning evaluation metrics for prediction accuracy, such as accuracy and sensitivity, are not suitable for analyzing image generation results. If the decision threshold is set too low, generated images may bear no meaningful relation to the real images, limiting evaluation to voxel-level comparisons. Conversely, if the threshold is set too high, even when there are significant differences between the generated and real images, the accuracy metric may still be high, which contradicts the actual situation. Therefore, after careful evaluation and multiple tests, this study chose SSIM, PSNR, and MI as the evaluation metrics for image translation quality.

---

## [Decision Letter · Decision Letter 2]

21 Jul 2025

RISNet: A Variable Multi-modal Image Feature Fusion Adversarial Neural Network for Generating Specific dMRI Images

PONE-D-25-02963R2

Dear Dr. hao,

We’re pleased to inform you that your manuscript has been judged scientifically suitable for publication and will be formally accepted for publication once it meets all outstanding technical requirements.

Kind regards,

Wang Zhan, Ph.D.

Academic Editor

PLOS ONE

Additional Editor Comments (optional):

Reviewers' comments:

Reviewer's Responses to Questions

**Comments to the Author**

1. If the authors have adequately addressed your comments raised in a previous round of review and you feel that this manuscript is now acceptable for publication, you may indicate that here to bypass the “Comments to the Author” section, enter your conflict of interest statement in the “Confidential to Editor” section, and submit your "Accept" recommendation.

Reviewer #1: All comments have been addressed

2. Is the manuscript technically sound, and do the data support the conclusions?

Reviewer #1: (No Response)

3. Has the statistical analysis been performed appropriately and rigorously? 

Reviewer #1: (No Response)

4. Have the authors made all data underlying the findings in their manuscript fully available?

Reviewer #1: (No Response)

5. Is the manuscript presented in an intelligible fashion and written in standard English?

Reviewer #1: (No Response)

6. Review Comments to the Author

Reviewer #1: (No Response)

7. PLOS authors have the option to publish the peer review history of their article (what does this mean?). If published, this will include your full peer review and any attached files.

Reviewer #1: **Yes: **Ghassan Abdul-Majeed

---

## [Editor Report · Acceptance letter]

PONE-D-25-02963R2

PLOS ONE

Dear Dr. Wang,

I'm pleased to inform you that your manuscript has been deemed suitable for publication in PLOS ONE. Congratulations! Your manuscript is now being handed over to our production team.

Kind regards,

on behalf of

Dr. Wang Zhan

Academic Editor

PLOS ONE